# Comparison of Growth Performance and Meat Quality Traits of Commercial Cross-Bred Pigs versus the Large Black Pig Breed

**DOI:** 10.3390/ani11010200

**Published:** 2021-01-15

**Authors:** Yongjie Wang, Keshari Thakali, Palika Morse, Sarah Shelby, Jinglong Chen, Jason Apple, Yan Huang

**Affiliations:** 1Department of Animal Science, Division of Agriculture, University of Arkansas, Fayetteville, AR 72701, USA; yw030@uark.edu (Y.W.); pdias@uark.edu (P.M.); sshelby@uark.edu (S.S.); 2Arkansas Children’s Nutrition Center, Department of Pediatrics, University of Arkansas for Medical Sciences, Little Rock, AR 72207, USA; KMThakali@uams.edu; 3Key Laboratory of Animal Physiology & Biochemistry, College of Veterinary Medicine, Nanjing Agricultural University, Nanjing 210095, China; 2017207005@njau.edu.cn; 4Department of Animal Science and Veterinary Technology, Texas A&M University, Kingsville, TX 78363, USA; Jason.Apple@tamuk.edu

**Keywords:** commercial cross-bred, purebred, meat quality, intramuscular fat, RNA-seq

## Abstract

**Simple Summary:**

Emphasis on improving meat quality is growing in the meat industry. Pigs under the same diet and environment could present with difference in meat quality if they have diverse genetic backgrounds. The Large Black pig is a British native pig breed which is famous for its hardness and resistance to extensive farming; compared with commercial cross-bred pigs, they have a slower growth rate. The aim of this study is to investigate the carcass and meat quality traits of Large Black pigs and commercial cross-bred pigs in relation to their transcriptome profiles, and consequently clarify the phenotypic and genotypic differences between these two groups of pigs. The results showed that Large Black pigs had greater intramuscular fat content than commercial cross-bred pigs, while the growth performance of commercial cross-bred pigs was better, and the transcriptomic differences between these two groups of pigs may be the cause of meat quality and growth performance variances. The outcome of the study provides new sights into the adoption of the Large Black breed in the pig industry.

**Abstract:**

The meat quality of different pig breeds is associated with their different muscle tissue physiological processes, which involves a large variety of genes related with muscle fat and energy metabolism. Understanding the differences of biological processes of muscle after slaughter is helpful to reveal the meat quality development of different breeds. Therefore, eight native Large Black pigs (BP), with high fat content in meat, and seven cross-bred commercial pigs (CP), which had a high feed efficiency with high lean meat, were used to investigate the differences in their meat quality and RNA transcriptomes. The average daily gain (ADG) and hot carcass weight (HCW) of CP were higher than BP, but the back-fat thickness of BP was higher than CP (*p* < 0.05). The CP had higher a* (redness) but lower h (hue angle) than BP (*p* < 0.05). The metmyoglobin (MMb) percentage of CP was higher (*p* < 0.05) than BP. The fat content and oxygen consumption of *longissimus dorsi* (LD) muscles in BP were higher (*p* < 0.05) than CP. BP had higher monounsaturated fatty acids (MUFA) content, but CP had higher polyunsaturated fatty acids (PUFA) content (*p* < 0.05). The RNA-seq data highlighted 201 genes differentially expressed between the two groups (corrected false discovery rate (FDR) *p* < 0.05), with 75 up-regulated and 126 down-regulated genes in BP compared with CP using the fold change (FC). The real-time PCR was used to validate the results of RNA-seq for eight genes, and the genes related to lipid and energy metabolism were highly expressed in BP (*p* < 0.05). Based on the results, BP had superior intramuscular fat content to CP, while the growth performance of CP was better, and the transcriptomic differences between these two groups of pigs may cause the meat quality and growth performance variance.

## 1. Introduction

Increasing the carcass weight and leanness of pigs has been a strong emphasis in production efficiency in swine breeding programs for many years. The researchers concentrated on improving pork production, such as the growth rate and feed conversion ratio, and they also aimed at decreasing carcass fat content and backfat thickness. However, pork quality has received a prime focus on pig production as consumers demand better meat quality, for example, tenderness, marbling, color, and water holding capacity (WHC) [1]. The interactive effects of pig breeds, environmental situations, pre-harvest management and post-harvest processes result in different meat qualities [2]. However, the effects of genotypes on meat quality are generally higher than external feed conditions [3]. It was reported that meat quality was also related to postmortem meat metabolism, and the oxygen consumption of muscle tissue is one of its metabolic phenotypes [4]. In addition, a large variety of genes related to both muscle structures and metabolic substances could affect the postmortem physiological developments inside the muscle cells. Comparing the transcriptome expression profile differences between different breeds could help us to understand the principles of genes related with meat quality and muscle biological processes.

Commercial cross-bred pigs (CP) are widely used in pig industry for its great feed efficiency and higher average daily gain (ADG), with high productivity of lean meat and less fat content. In contrast, the Large Black pigs (BP) are British native pigs with a long and deep-bodied shape. They are well known for their high fat content and high meat quality characteristics [5,6].

The aim of this experiment was to investigate the carcass and meat quality traits of Large Black pigs (BP) and commercial cross-bred pigs (CP) in relation to their transcriptome profiles, and consequently clarify the phenotypic and transcriptomic differences between these two groups of pigs. The outcome of the study provided new sights into the adoption of the Large Black breed in the pig industry. In order to investigate the differences of gene expression between BP and CP, the RNA-seq methodology, which is widely used to provide a comprehension transcriptomic profile of analyzed tissue, has been applied [7,8]. Based on RNA-seq results, functional analysis of gene ontology (GO) and biological processes were used to highlight enriched relevant biological pathways [9,10]. In addition, the real-time PCR was used to support the RNA-seq data.

## 2. Material and Methods

### 2.1. Animals

The University of Arkansas’s Institutional Animal Care and Use Committee approved all procedures of the experiment involving animals during the study (ethical approval code: 18,000). Eight Large Black pigs and seven commercial cross-bred pigs (PIC 29 dam × PIC 380 boar) were allocated to the BP group and CP group, and their initial mean body weights were tabulated (23.31 ± 1.93 kg for the BP group and 18.82 ± 1.41 kg for the CP group). They were fed ad libitum and kept individually in digestibility pens for 101 days. The final body weight of the CP group was determined at the end of the 101-day experiment. However, due to low growth performance of the BP, the average body weight of BP group was 115.4 kg, lower than the average market live weight, which is 127.9 kg [11]. The BP group was kept for 108 days. The differences between the initial and final body weight was used to calculate the ADG. Animals were fasted 12 h prior to slaughter, with access to water. Then, they were rendered unconscious by a nonpenetration stunning method.

### 2.2. Carcass Characteristics and Sampling

Immediately after stunning and completion of exsanguination, hot carcass weight (HCW) and backfat thickness (midline, between the 4th and 5th lumbar vertebra level) were determined. Then, muscle samples of *longissimus dorsi* (LD) of the left carcass was removed, and the samples were snap frozen at −80 °C in liquid nitrogen for RNA isolation process. The carcasses were chilled at −9 °C for two hours and then kept at 4 °C for 24 h. Another piece from the LD muscle was obtained (between the 12th and 13th rib) and transported under refrigeration until being processed for meat quality. The muscles were cut into small pieces, vacuum-packed, and stored at −20 °C before analyzing the fatty acid profile.

### 2.3. Meat Quality

The intramuscular fat (IMF) content of the *longissimus dorsi* muscle was determined by using Soxhlet apparatus to extract ether without prior acid hydrolysis [12]. A 100 g thick slice cut from the LD muscle was placed into a polypropylene bag and then stored in a vacuum package for 24 h at 4 °C, and the weight differences of samples were regarded as drip loss, showed as percentages [13]. In order to ensure stable data of color measurements, samples were bloomed for 20 min before further analysis. The Lightness (L), red to green (a*), and yellow to blue (b*) color values were determined by a CR-400 Chroma Meter. Then, the hue angle (h = arctan (b*/a*)) and chroma (C* = ((a*)^2^ + (b*)^2^)^0.5^) were calculated using the L, a*, and b* values.

### 2.4. Fatty Acid Composition

The fatty acid composition of LD muscle IMF was determined by fat extraction [14]. Ten grams of minced meat were homogenized at 3000 rpm for 1.5 min by UltraTurrax using 0.003% butylhydroxytoluene (BHT) with 200 ml Folch solution (chloroform–methanol mix 2:1). After paper-filtering (Whatman No. 1) the homogenized liquid, Folch solution (50 mL) was added. After filtering, the solution was poured out into a decantation infundibulum and mixed with 8% sodium chloride (80 mL) for 24 h. The solvent, collected from the lipidic phase, was evaporated. After evaporation, the fatty acid composition was analyzed using gas chromatography (Agilent 6890 N Network GC System). As a carrier gas, helium was used at a division ratio of 1:50 with a 3.2 ml per minute flow rate. The undecanoic acid methyl ester was used as an internal standard to quantify the methyl esters of fatty acids.

### 2.5. Oxygen Consumption and Myoglobin Calculation

Approximately 20 mg of tissue from the *longissimus dorsi* (LD) muscle from each sample was put into a respiration buffer (1.1 mM sodium pyruvate, 25 mM glucose in PBS and 2% BSA). The oxygen consumption rate (OCR) of every sample was measured by an Orion Dissolved Oxygen Platform (Scientific) about 25 min and repeated 3 times. A full reflectance spectral analysis was also taken on each muscle, and wavelength ratios were used to calculate relative deoxymyoglobin (474:525 nm), oxymyoglobin (610:525 nm), and metmyoglobin (572:525 nm) concentrations [15].

### 2.6. RNA Extraction and cDNA Synthesis

The total RNA of *longissimus dorsi* muscles was extracted by TRIzol Reagent (Invitrogen, Carlsbad, CA, USA) with mechanic homogenization by using the Precellys Evolution homogenizer (Bertin Technologies, Rockville, MD, USA). The total RNA, extracted from the muscle, was then treated with DNase I (Promega, Madison, WI, USA) to eliminate the contamination of genomic DNA, following the directions of the manufacturer. The concentration of total RNA was assessed by NanoDrop (Agilent Technologies, CA, USA). The same RNA extracts were used both for real-time qPCR and RNA sequencing later. Reverse transcription of the decontaminated RNA samples was performed using Takara PrimeScript^TM^ RT reagent kit, and then prepared for cDNA synthesis according to the instructions of Takara PrimeScript^TM^ RT.

### 2.7. Statistical Analysis

All the data analysis was performed using SPSS version 19.0 software (SPSS Inc., Chicago, IL, USA). The differences of mean values of these two groups were carried out using *t*-tests of independent samples. The results are given as means and SD in the text, and the differences were considered significant when *p* < 0.05.

### 2.8. Real-Time qPCR

According to the recorded sequences showed in GenBank, the primers of *SLC26A7, TKTL2, ACBD7, THRSP, SLPI, FADS1, ACSL6, FOS* and *GAPDH* were designed using Oligo 6.0 Software, and they were designed to span the introns to make sure the correct cDNAs are amplified (Table 1). GAPDH gene has been widely used as a housekeeping gene normalizer [16], and was thus selected as a reference in the present study. The cDNA was used to perform real-time PCR in order to obtain the expression levels of *SLC26A7, TKTL2, ACBD7, THRSP, SLPI, FADS1, ACSL6, FOS.* Real-time-qPCR was performed using 15 μL reaction system: 7.5 μL 2 × Real Master Mix; 0.75 μL upstream and 0.75 downstream primer (10 pmol/L); 3 μL cDNA; and 3 μL water. The reaction liquid was added on iCycler IQ^5^ (Bio-Rad, USA). Technical duplicates were applied for the qPCR analysis. The PCR procedure was denaturing the DNA at 95 °C for 3 min, followed by 40 cycles of denaturation at 95 °C for 15 s, and annealing/extension at 55 °C for 30 s. Relative expression levels were normalized to the GAPDH gene and expressed as fold change [17]. The software CFX manager (Bio-Rad, USA) was used to process Ct values, and the delta-delta Ct method was used to calculate the fold change.

### 2.9. RNA Sequencing and Functional Analysis

The RNA was isolated from the *longissimus dorsi* muscle using Trizol reagent as described above. Three micrograms of total RNA were used to isolate poly-A RNA using the Dynabeads mRNA Direct Kit (Thermo Fisher Scientific, Waltham, MA, USA) [18]. Three biological pools were made for each breed of pig. In the commercial cross-bred pigs, three pigs, two pigs, and two pigs were pooled. In the Large Black pig group, three, three, and two pigs were pooled. The RNA fragmentation, library prep, and adapter ligation were performed using the NEBNext Ultra Directional Library Prep kit for for Illumina (New England Biolabs, Ipswich, MA, USA). Library size and quality were assessed using the High Sensitivity DNA Analysis Kit for the Agilent Bioanalyzer (Agilent, Santa Clara, CA, USA) and library concentration was determined using the Qubit dsDNA HS Assay Kit (Thermo Fisher Scientific). Single-read 75-bp sequencing of libraries were performed using a NextSeq500 (Illumina, San Diego, CA, USA). The FastQC app in BaseSpace was used to assess read quality and the FASTQ Toolkit was used for adapter trimming, base trimming low quality reads at the 3′ end, and filtering using a minimum mean quality score of 30 (Illumina) (http://basespace.illumina.com/apps/). Alignment was performed using the Star Alignment app in BaseSpace and aligned to the pig genome (sus scrofa 11.1_v91) (Appendix A) [19]. The resulting Bam files were uploaded into SeqMonk (v.1.37.1, http://www.bioinformatics.babraham.ac.uk/projects/seqmonk/) for differential expression analysis using the EdgeR package in SeqMonk (*p* < 0.05 after FD correction) [20]. The lists of differentially expressed genes were further analyzed for GO of biological function enrichment using the Database for Annotation, Visualization and Integrated Discovery (DAVID) tool [21,22].

## 3. Results and Discussion

### 3.1. Growth Performance and Carcass Measurements

The growth performances in terms of initial and final body weight and the quality of carcass are presented in Table 2. The initial body weights of CP and BP groups showed no significant differences (*p* > 0.05). However, the CP group had higher final body weight (*p* < 0.05), ADG (*p* < 0.05) and hot carcass weight (*p* < 0.01) compared with the BP group. According to Bessa et al. [23], the fat deposition of the Large Black pig is higher than the crossbreed pig. Correspondingly, the BP group showed a strong fat deposition ability with significantly higher (*p* < 0.01) back fat thickness values compared to CP.

### 3.2. Meat Quality

Meat quality traits of CP and BP groups are summarized in Table 3. Drip loss is one of the important characteristics of water holding capacity (WHC). There was no significant difference shown in drip loss between CP and BP (*p* > 0.05). For the meat color, L* indicates lightness, a* relates to the red/green coordinate, b* shows the yellow/blue coordinate, C* means chroma, and h is the hue angle [24]. There were no significant differences in L*, b* and C* between these two groups (*p* > 0.05). In contrast, the a* value of CP was higher (*p* < 0.01) than BP, but the h value of BP was higher (*p* < 0.05) than CP.

The IMF content of these two breeds was significantly different (*p* < 0.05), meaning that the IMF of BP is higher than that of CP pigs. However, the different fat content did not affect the drip loss. Watanabe et al. [25] pointed out that pork IMF content was not correlated with drip loss, and our result had the same trend. The fat content of CP is similar with other reports about white pig breeds [26,27], and the fat deposition of BP is much higher than these commercial pigs. Research comparing Iberian pigs and commercial cross-bred pigs showed a strong correlation between IMF and backfat thickness [28]. Our data showed higher IMF and backfat deposition in the Large Black pigs, which demonstrated the similar fat deposition patterns in local native pigs which is different from commercial cross-bred pigs.

### 3.3. Oxygen Consumption

The OCR is positively correlated with mitochondrial concentration [29]. In addition, mitochondria present in the postmortem muscle can remain active and impact on the meat color through oxygen consumption [30]. The OCR of BP was higher than that in CP (*p* < 0.01; Figure 1). It was reported that muscles with more β-red fibers tend to have higher OCR than muscles with more α-white fibers [31]. However, the relationship between oxygen consumption and meat quality needs to be further evaluated to understand color variations between the two groups of pigs. A study of the muscle fiber type of Korean native pig reported a higher proportion of type I muscle fibers than that in commercial cross-bred pigs [32]. The type I slow twitch fiber has a higher mitochondria content for oxidative phosphorylation. This report was consistent with what we found; the local native pigs had higher mitochondrial metabolism correlated with the higher content of type I muscle fiber.

### 3.4. Myoglobin Calculation

The color of meat after slaughter is primarily affected by myoglobin, which is a sarcoplasmic heme protein [33]. This protein can exist as deoxymyoglobin (DMb), oxymyoglobin (OMb), and metmyoglobin (MMb) in fresh meat [30]. Our results showed differences in the percentage of myoglobin between CP and BP breeds (Figure 2). The MMb (metmyoglobin) percentage of CP was higher (*p* < 0.05) than BP, but there were no significant differences between CP and BP in DMb (deoxymyoglobin) and OMb (oxymyoglobin). The chemistry and functions of myoglobin in live muscles and meat may be different, but the functions of myoglobin as the oxygen binder and oxygen deliverer to maintain the physiological functions of mitochondria continues normally [34]. In addition, myoglobin is related to red meat color, which is a main meat purchasing factor for consumers [35].

### 3.5. Fatty Acid Composition

The fatty acid compositions of LD muscles from CP and BP breeds are listed in Table 4. There was a significant fatty acid composition difference between CP and BP in most comparisons (*p* < 0.05). The total polyunsaturated fatty acid contents of CP were higher (*p* < 0.05) than BP, such as C18:2n-6, C18:3n-3, C20:2, C20:3n-6, C20:4n-6, and C22:5. However, BP had higher total monounsaturated fatty acid content along with higher C:10, C:20, C18:1, and C20:0 contents compared to CP. The overall saturated and unsaturated fatty acids contents of CP and BP were not significantly different (*p* > 0.05).

The fatty acid composition has a strong relationship with the IMF content and backfat thickness [36]. It was reported that intramuscular and backfat would increase the percentage of saturated, especially monounsaturated fatty acids, and the data of our experiment are in agreement with these findings [37]. The fatty acid composition variations in CP and BP are probably attributed to the difference in fat deposition between these two groups of pigs. The SFA level of intramuscular fat in the loin has negative correlations with meat sensory qualities, such as acid flavor [38]. However, there were no significant differences of SFA between BP and CP. It was reported that monounsaturated fatty acids are the major fatty acid component in the Mediterranean diet, of which benefits include lowering the risk of cardiovascular disease [39]. The IMF content in BP LD muscle is about 1.6 times higher than CP (10.02 ± 1.20:6.24 ± 1.53, *p* < 0.05. Table 2), which might indicate that BP pork products have more health value with a higher content of monounsaturated fatty acids.

There are limitations in the results of growth performance, meat color, and fatty acid profile due to the small sample numbers. These data may not reflect the obtained results to a breed-wide interpretation. Future study will utilize a bigger range of the sample numbers.

### 3.6. Differentially Expressed Gene Analysis

High-throughput sequencing is a powerful way to identify the differences in gene expression, which was used in the study of different breeds to compare the difference of gene expressions related with meat quality [40]. By comparing LD muscle transcriptome differences between BP and CP, we found that there was a total of 363 differentially expressed genes (Appendix A) found between these two breeds, in which 201 were highly differentially expressed (log2 fold change ≥1 or ≤ −1; *p* < 0.05). Full details of gene name, description, identification, and fold change (FC) are reported in the Appendix A. Compared with CP, BP had 75 up-regulated and 126 down-regulated genes (Figure 3). The functional category of these 201 differentially expressed genes, of which 75 were highly expressed in BP and 126 were highly expressed in CP, were determined by querying associated gene ontologies, and they were classified into biological process, cellular component, and molecular function (Figure 4, Figure 5 and Figure 6), by using DAVID bioinformatic resources. GO analysis showed the functional enrichment of these differentially expressed genes, and the different genes expressed may be the cause of diversities in carcass characteristics and meat quality between these two groups of pigs. For biological processes, five highly expressed genes in BP were related with fat cell differentiation (Figure 4). This might have led to a higher IMF content of LD muscle in BP, compared with CP. In molecular functions, we found that six genes, which were highly expressed in CP, were responsible for oxidoreductase activity, and these genes’ higher expression may cause the different OCR of the LD muscle between BP and CP.

### 3.7. Gene Expression

In order to verify our RNA-seq expression profile data, we utilized real-time quantitative RT-PCR to determine eight genes related to lipid deposition or metabolism, and the results (Figure 7) partly validated the transcriptome profiles of RNA-seq. Results are presented as numerical relative gene expression values (using *GAPDH* as a housekeeping calibrator). The expression of lipid and energy metabolic related genes (*TKTL2*, *ACBD7*, *ACSL6,* and *FOS*) were higher in CP than in BP (*p* < 0.05). In addition, CP exhibited higher (*p* < 0.05) mRNA abundance of the *THRSP* gene, which is related to medium-length fatty acid chains [41]. However, the gene expression of SLPI which is involved in epithelial immunity was higher in BP (*p* < 0.05) and is consistent with the characteristic of stronger environmental tolerance in BP as a native pig breed compared to CP [42]. The genes related to energy metabolism and unsaturated fatty acids composition (*SLC26A7* and *FADS1*) were more highly expressed in CP. Pigs of different breeds always have different conditions of metabolism and lipid deposition. *SLC26A7* is strongly related to energy metabolism [43]. *TKTL2* is related to lipid transport and metabolism, and the genetic distance increased from local pig breeds after selective seep caused by the selection of energy-rich pork production [44]. *ACBD7* is one of the gene families related to intracellular lipid-binding proteins [45]. The *FADS1* gene is associated with the unsaturated fatty acid composition [46]. It has been reported that *ACSL6* contributes to lipid synthesis [47]. Reiner et al. [48] reported that *FOS* is an important gene correlated with skeletal muscle fiber and metabolism. Above all, the differential of gene expression related to lipid and energy metabolism might cause the different fat deposition ability of CP and BP.

## 4. Conclusions

In conclusion, the growth performance of CP was higher than BP, but the intramuscular fat content of BP was higher than CP, which indicates higher meat quality in the BP group. The meat oxygen consumption rate of LD muscle in the BP group was also high; therefore, the Large Black pigs have higher metabolic rate and possibly more type I muscle fiber. The RNA-seq and gene expression data qualitatively and quantitatively proved that there were differences in intramuscular transcriptome profiles of these two groups of pigs. Comparing BP with CP, 201 significantly differentially expressed genes in the LD muscle were identified. The analysis of the genes between the two groups of pigs may provide a novel strategy in swine genetic selection for both high growth performance and high meat quality, especially concerning marbling and flavor.

## Figures and Tables

**Figure 1 animals-11-00200-f001:**
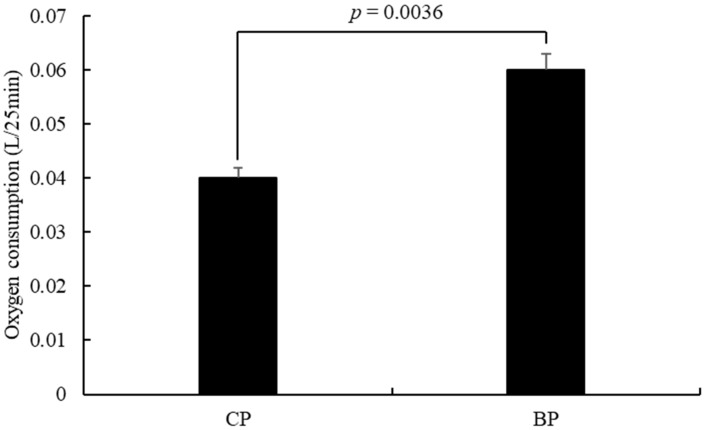
Effect of breeds on the oxygen consumption (*longissimus dorsi*) (CP: cross-bred commercial pigs; BP: Large Black pigs).

**Figure 2 animals-11-00200-f002:**
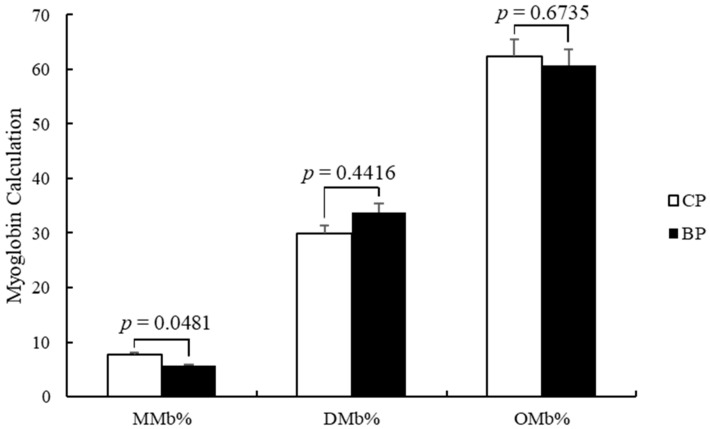
Effect of breeds on the myoglobin calculation (*longissimus dorsi*) (CP: cross-bred commercial pigs, BP: Large Black pigs; MMb: Metmyoglobin, DMb: Deoxymyoglobin, OMb: Oxymyoglobin).

**Figure 3 animals-11-00200-f003:**
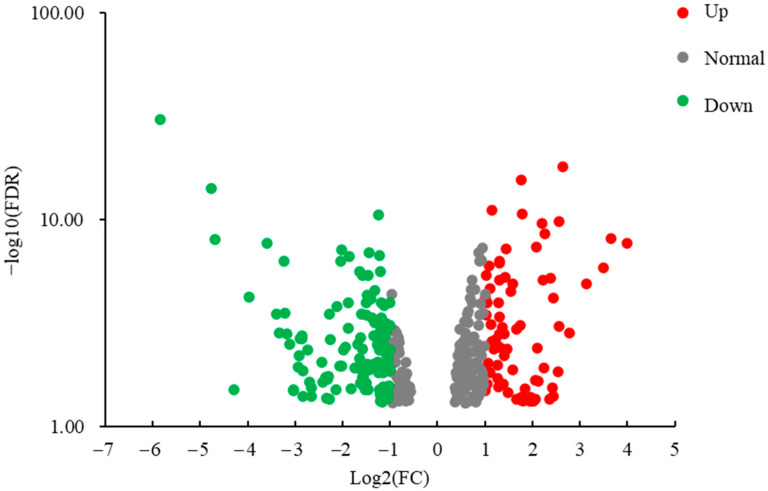
Volcano figure of RNA-seq (BP vs. CP; CP: cross-bred commercial pigs, BP: Large Black pigs; Up: up-regulated in BP compared with CP; Normal: no significant differences; Down: down-regulated in BP compared with CP).

**Figure 4 animals-11-00200-f004:**
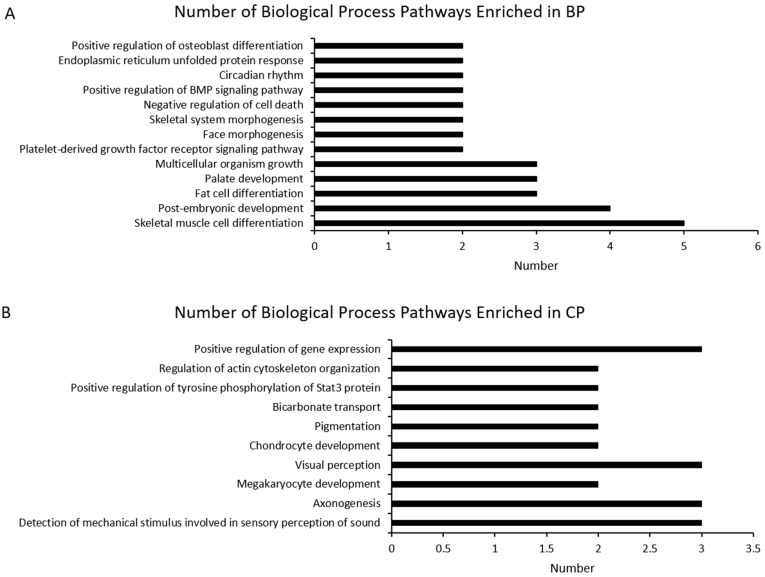
Gene ontology (GO) analysis of differentially expressed genes highly expressed in BP (**A**) and CP (**B**). The identified differentially expressed genes were classified into biological processes. The numbers of genes in the GO term is shown above (BP vs. CP; CP: cross-bred commercial pigs, BP: Large Black pigs).

**Figure 5 animals-11-00200-f005:**
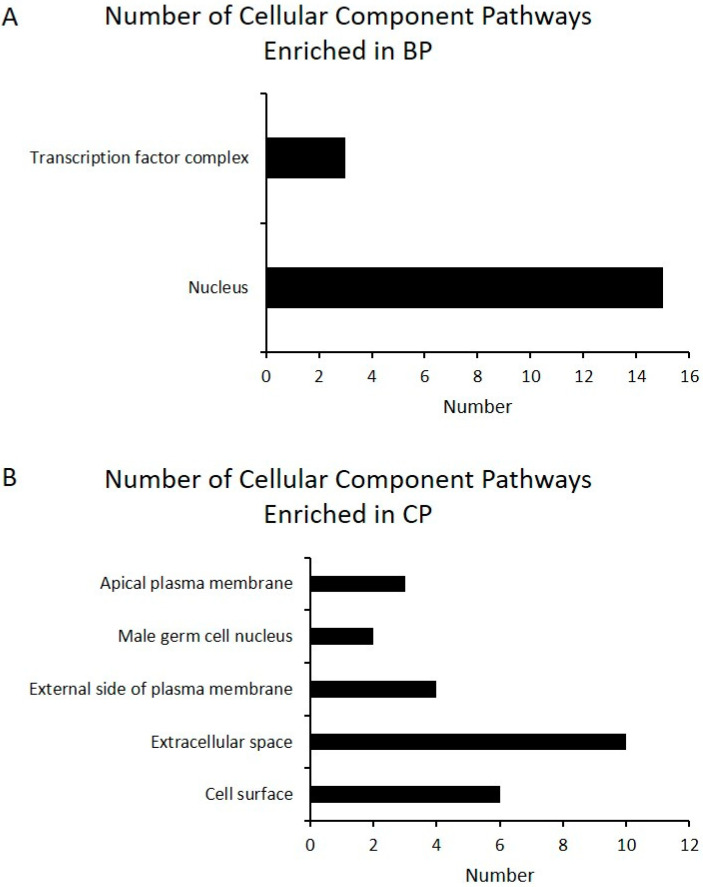
Gene ontology (GO) analysis of differentially expressed genes highly expressed in BP (**A**) and CP (**B**). The identified differentially expressed genes were classified into cellular components. The number of genes in the GO term is shown above (BP vs. CP; CP: cross-bred commercial pigs, BP: Large Black pigs).

**Figure 6 animals-11-00200-f006:**
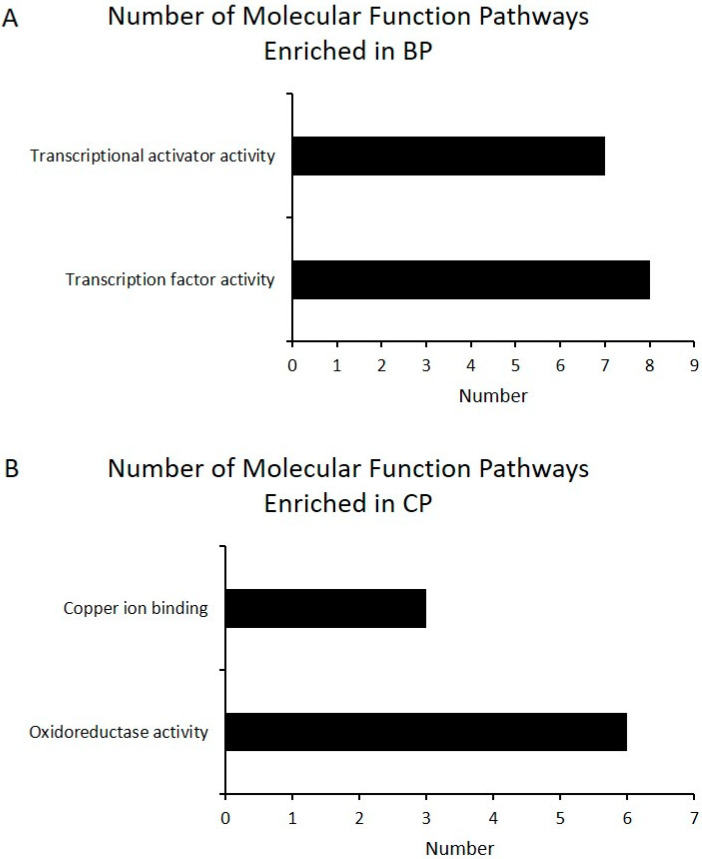
Gene ontology (GO) analysis of differentially expressed genes highly expressed in BP (**A**) and CP (**B**). The identified differentially expressed genes were classified into molecular functions. The number of genes in the GO term is shown above (BP vs. CP; CP: cross-bred commercial pigs, BP: Large Black pigs).

**Figure 7 animals-11-00200-f007:**
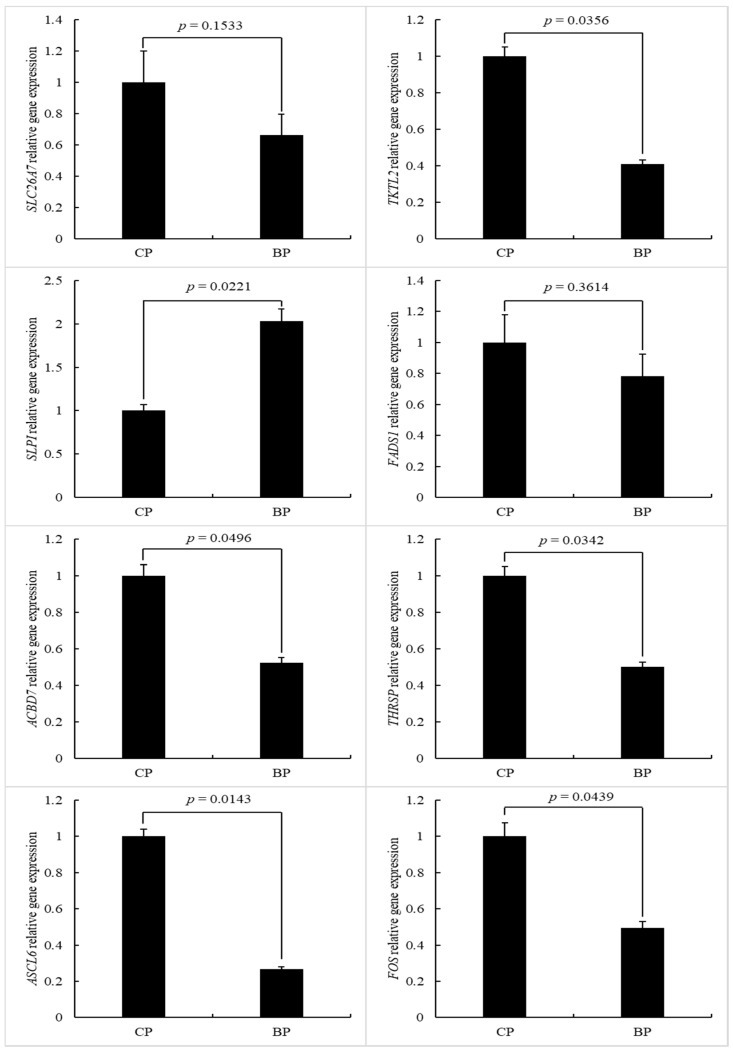
Effect of breeds on the relative gene expression (BP vs. CP; CP: cross-bred commercial pigs, BP: Large Black pigs).

**Table 1 animals-11-00200-t001:** Primer information for genes chosen for confirmation of expression using quantitative real-time PCR.

Gene	Direction	Primer Sequence	GenBank Accession No.	Binding Positions	Product Length (bp)
*GAPDH*	Forward Reverse	5′-TCGGAGTGAACGGATTTGGC-3′ 5′-TGACAAGCTTCCCGTTCTCC-3′	NM_00120659.1	111 299	189
*SLC26A7*	Forward Reverse	5′-GAAAATGCCAGCAATCAGCCA-3′ 5′-AGGGCCACAGTTCCCATTG-3′	XM_021089130.1	1837 1938	102
*TKTL2*	Forward Reverse	5′-CTGGCCTTTGCATCCCACTA-3′ 5′-GTATCCATGCAGTGCGCAAG-3′	XM_013978706.2	373 476	104
*ACBD7*	Forward Reverse	5′-GGAAGATGCCATGAGTGCCT-3′ 5′-CTGAGGGCTTCAAAAGGCAAA-3′	XM_003357745.4	259 362	104
*THRSP*	Forward Reverse	5′-GTAGCCTCGGACTCTAGGCA-3′ 5′-CTGCAGGTCCAGGTCTTTCT-3′	NM_001244376.1	486 599	114
*SLPI*	Forward Reverse	5′-CAAGTGCACAAGTGACTGGC-3′ 5′-GGCCATAGACCACTGGACAC-3′	NM_213870.1	139 275	137
*FADS1*	Forward Reverse	5′-GTCACTGCCTGGCTCATTCT-3′ 5′-AGGTGGTTCCACGTAGAGGT-3′	NM_001113041.1	433 587	155
*ACSL6*	Forward Reverse	5′-GAATACGGGCACTCTCTGGC-3′ 5′-CCTAGGACCCCAGTTTGCAG-3′	XM_021084743.1	44 173	130
*FOS*	Forward Reverse	5′-GACTGCTATCTCGACCAGCC-3′ 5′-CTGGCATGGTCTTCACGACT-3′	NM_001123113.1	192 349	158

**Table 2 animals-11-00200-t002:** Effect of breeds on the carcass characteristics.

Parameters	CP ^a^		BP		*p*-Value
	Mean	sd	Mean	sd	
Initial body weight (kg)	18.82	1.41	23.31	1.93	0.1097
Final body weight (kg)	130.04	8.16	121.17	2.80	0.0341
ADG ^b^ (kg)	1.10	0.05	0.91	0.01	0.0452
Hot carcass weight (kg)	77.95	9.81	63.63	0.95	0.0060
Back fat thickness (cm)	0.72	0.07	1.42	0.22	0.0004

^a^ CP: cross-bred commercial pigs, BP: Large Black pigs; ^b^ ADG: average daily gain.

**Table 3 animals-11-00200-t003:** Effect of breeds on the meat quality of pork (*longissimus dorsi*).

Parameters	CP ^a^		BP		*p*-Value
	Mean	sd	Mean	sd	
Drip loss (%)	3.38	0.11	3.41	0.17	0.9571
L* ^b^	57.78	4.52	60.24	2.43	0.3072
a*	16.72	1.73	14.58	1.09	0.0063
b*	14.31	1.68	13.73	0.67	0.4192
C*	22.02	2.28	19.99	1.09	0.0951
h	40.49	2.13	43.47	1.64	0.0176
Proximal composition:					
Total fat (%)	6.24	1.53	10.02	1.20	0.0431

^a^ CP: cross-bred commercial pigs, BP: Large Black pigs; ^b^ L*: lightness; a*: red–green; b*: yellow–blue; C*: bright–dull; h: hue.

**Table 4 animals-11-00200-t004:** Effect of breeds on the fatty acid profile of intramuscular fat (*Longissimus dorsi*).

	CP ^a^		BP		*p*-Value
	Mean	sd	Mean	sd	
C10:0	0.08	0.01	0.10	0.01	0.0074
C12:0	0.06	0.01	0.07	0.01	0.2895
C14:0	1.14	0.17	1.20	0.08	0.5063
C16:0	23.76	0.66	24.21	0.73	0.3495
C16:1	3.06	0.46	3.32	2.41	0.2189
C17:0	0.23	0.03	0.24	0.02	0.5706
C18:0	11.74	0.79	11.95	0.73	0.4128
C18:1	41.81	3.20	46.21	1.05	0.0153
C18:2n-6	12.67	2.74	8.74	1.40	0.0352
C18:3n-3	0.27	0.02	0.23	0.04	0.0496
C20:0	0.13	0.01	0.19	0.02	0.0013
C20:1	0.51	0.10	0.70	0.07	0.0007
C20:2	0.29	0.04	0.23	0.02	0.0061
C20:3n-6	0.35	0.11	0.22	0.05	0.0085
C20:4n-6	2.82	0.87	1.53	0.48	0.0032
C22:5	0.28	0.09	0.14	0.04	0.0162
SFA ^b^	37.07	1.09	37.96	1.40	0.7946
UFA	62.04	1.05	61.25	1.34	0.5138
MUFA	45.32	3.61	50.18	1.21	0.0006
PUFA	3.75	1.06	2.11	0.55	0.0004
n6/n3	27.59	3.63	28.35	3.08	0.0901

^a^ CP: cross-bred commercial pigs, BP: Large Black pigs; ^b^ SFA: saturated fatty acid; UFA: unsaturated fatty acid; MUFA: monounsaturated fatty acid; PUFA: polyunsaturated fatty acid; n6/n3: the ratio of n-6 fatty acids to n-3 fatty acids.

## Data Availability

The datasets used and/or analyzed during the current study are available from the corresponding author on reasonable request.

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
