# Peer review of "Comparison of Growth Performance and Meat Quality Traits of Commercial Cross-Bred Pigs versus the Large Black Pig Breed"

_animals, 2021, doi:10.3390/ani11010200_

Round 1

Reviewer 1 Report

In Materials and Methods section it would be appropriate to indicate the breed composition of cross-bred pigs (if it is known for authors). The authors incorrectly use the term „breed“ throughout the manuscript, because commercial cross-bred or hybrid pigs are not the breed. Therefore, for comparisons between Large Black and commercial cross-bred pigs it is inaccurate to use „between these two breeds“ (Lines 22,24,39, 43, 69, 200, 236, 258, 309, 311). It should be corrected.

The authors also unnecessarily and too often use the term „domestic,“ particularly describing British Large Black  pigs. Commercial cross-bred pigs are also domestic pigs. Although their origin was not indicated in the manuscript, most likely commercial cross-bred pigs were obtained from the cross of modern pig breeds without incorporation of wild boar and feral pigs. It should be corrected.

The statement „Based on the results, BP had superior general meat quality to CP“ should be clarified on which results (evaluated parameters) such conclusion was made (Lines 41-42). Section Conclusions also should be clarified similarly.

Since BP pigs had lower percentage of PUFA in IMF, n-6/n-3 PUFA was not lower compared to CP pigs (Lines 241-242), also lipid quality indices, such as atherogenic, thrombogenic and peroxidizability indexes, and the hypocholesterolemic/hypercholesterolemic ratio were not calculated, the statement “which might indicate that BP pork products have higher healthy value“(Lines 241-242) made on the basis of only higher MUFA percentage should be supported with more detailed explanation.

The markings on the graph and their explanations in the footnote of Figure 3 should coincide.

Keywords: compound keywords, such as  „ Commercial cross-bred pigs; British Large Black pigs“are not only too long, but also have narrow meanings. They could be replaced by  „pigs, breed, commercial cross-bred, purebred.“

Author Response

Thanks for the comments.

  1. In Materials and Methods section it would be appropriate to indicate the breed composition of cross-bred pigs (if it is known for authors). The authors incorrectly use the term “breed” throughout the manuscript, because commercial cross-bred or hybrid pigs are not the breed. Therefore, for comparisons between Large Black and commercial cross-bred pigs it is inaccurate to use “between these two breeds” (Lines 22,24,39, 43, 69, 200, 236, 258, 309, 311). It should be corrected.

Response: Thanks for the advice. We do not know the breed composition of commercial cross-bred pigs. The “breed” has already been corrected.

  1. The authors also unnecessarily and too often use the term “domestic”, particularly describing British Large Black  pigs. Commercial cross-bred pigs are also domestic pigs. Although their origin was not indicated in the manuscript, most likely commercial cross-bred pigs were obtained from the cross of modern pig breeds without incorporation of wild boar and feral pigs. It should be corrected.

Response: The “domestic” has already been changed to native.

  1. The statement „Based on the results, BP had superior general meat quality to CP“ should be clarified on which results (evaluated parameters) such conclusion was made (Lines 41-42). Section Conclusions also should be clarified similarly.

Response: This section has been revised.

  1. Since BP pigs had lower percentage of PUFA in IMF, n-6/n-3 PUFA was not lower compared to CP pigs (Lines 241-242), also lipid quality indices, such as atherogenic, thrombogenic and peroxidizability indexes, and the hypocholesterolemic/hypercholesterolemic ratio were not calculated, the statement “which might indicate that BP pork products have higher healthy value“(Lines 241-242) made on the basis of only higher MUFA percentage should be supported with more detailed explanation.

Response: The explanation has been added.

  1. The markings on the graph and their explanations in the footnote of Figure 3 should coincide.

Response: Sorry, we didn’t find the noncoincidence. Please instruct more detailly. We moved the legend to a upper place that makes people easy to distinguish it from the figure.

  1. Keywords: compound keywords, such as  “ Commercial cross-bred pigs; British Large Black pigs” are not only too long, but also have narrow meanings. They could be replaced by  "pigs, breed, commercial cross-bred, purebred.“

Response: The key words have already been changed.

Reviewer 2 Report

Line 66: Content

Line 72: Freshly used seems like odd verbage. Do you mean it is the the newest/best technology?

Line 75: rapid advances

Line 105: Re-word "A 2 cm 100g thick"

Line 127: missing words

PCR: were primers intron spanning, do you have efficiency on the primers or other criteria one should use to determine the efficacy of primers?

Line 197: Change than to that

Line 240 and 241: Maybe you'll address below but is a 1.6 fold increase in monounsaturated fats a relevant contribution to the diet? Many times we've seen 2-3 fold increase in omega 3's in beef but they are not nutritionally relevant. 

Conclusion: There needs to be a conclusion. Differentially expressed genes were identified is a result, not a conclusion. What does is mean? How can this knowledge be used? Will improvements in meat quality going to help producer profitability? 

Author Response

Thanks for your comments.

  1. Line 66: Content

Response: It has been changed.

  1. Line 72: Freshly used seems like odd verbage. Do you mean it is the the newest/best technology?

Response: The “freshly” has been changed to widely.

  1. Line 75: rapid advances

Response: It has been changed.

  1. Line 105: Re-word "A 2 cm 100g thick"

Response: It has been changed to “A 100g thick”

  1. Line 127: missing words

Response: It has been revised.

  1. PCR: were primers intron spanning, do you have efficiency on the primers or other criteria one should use to determine the efficacy of primers?

Response: During designing the primers, we selected the primers based on the second structure such as hairpins and dimers. Standard curve and melting curve were applied to secure the efficiency of primers.

  1. Line 197: Change than to that

Response: It has been changed.

  1. Line 240 and 241: Maybe you'll address below but is a 1.6 fold increase in monounsaturated fats a relevant contribution to the diet? Many times we've seen 2-3 fold increase in omega 3's in beef but they are not nutritionally relevant.

Response: I mentioned in the article that a higher level of monounsaturated fatty acids could decrease the risk of cardiovascular disease, so it might be healthier with a higher content of monounsaturated fatty acids.

  1. Conclusion: There needs to be a conclusion. Differentially expressed genes were identified is a result, not a conclusion. What does is mean? How can this knowledge be used? Will improvements in meat quality going to help producer profitability?

Response: Sorry for making you confused about the Conclusion part. It is at the end of this article. The significance of this study has been added at the end of the conclusion. Thank you!

Reviewer 3 Report

The authors of the Manuscript entitled “Comparison of Growth Performance and Meat 2 Quality Traits of Commercial Cross-bred Pig versus Large Black Pig Breeds” report differences at both phenotype and transcriptomic level for a cohort of animals belonging to two pig breeds with commercial interest. Although comparing widely used commercial crossbred pigs with alternative less exploited local breeds is a relevant topic for the meat industry, this Manuscript reflects several flaws in study design, data management and results interpretation that make the current Manuscript almost unsuitable for publication.

First, I strongly suggest the authors to thoroughly revise the grammatical correctness and English usage across the text, as there are several easily avoidable errors in the use of English to report the results.

Second, phenotype-based comparison with such limited sample of animals is far from accurate and representative of the actual phenotype distribution across the population, and inferring significant differences with such limited N is quite risky, at best.

Third, there is a visible and traceable lack of knowledge in the treatment and interpretation of RNA-seq data results. Bioinformatic analyses are poorly described in Methods, and equally poorly interpreted and reported in Results and Discussion. I strongly suggest to thoroughly revise the extensive collection of high-quality RNA-seq studies reported in the literature, prior to manage and report RNA-seq data. Easily avoidable errors included in the Manuscript could be corrected if authors had a proper understanding of the intricacies of RNA-seq data analysis.

Given such concerns, I propose a thorough and proper correction of the study design and interpretation of the obtained results, prior to submitting the Manuscript to dedicated peer-review process in high-quality journals in the Animal Science field. I therefore propose this Manuscript as not suitable for publication and recommend Major Revision. However, given the extensive revision needed, I strongly suggest to authors that they reconsider the quality and correctness of their study. Please carefully reanalyze your data following good practices for performing and reporting RNA-seq and RT-qPCR analyses.

Following lines give more detail on a per-line basis:

Lines 14-26: This section contains several flawed grammatical constructions and repetitions which should be fixed and reoriented. I suggest to rewrite and carefully consider the general claims that are put throughout the Manuscript.

Lines 27-44: While I strongly suggest the authors not to overcrowd the abstract with acronyms, the use of these and their previous definition must be generalized for all of them used. For instance, the meaning of a*, h, SFA, MUFA and PUFA are not detailed. Please either do not use acronyms in the abstract, or apply a constant stable rule to define their corresponding meaning prior to their inclusion in the text.

Lines 45-46: “Breeds” and “Gene Expression” are probably useless and not meaningful as keywords given the content of the Manuscript.

Line 54: Please introduce a grammatically correct form to introduce Dransfield et al. reference. As put in the text, it is grammatically linked to WHC, which is obviously not the case.

Lines 56-57:  Please remove “eating” for defining meat quality. Also, the grammatically correct form of this sentence would be: “However, the effects of genotype on meat quality are generally higher than external feed conditions”.

Line 57-58: Please rephrase this sentence.

Line 71-80: At present days, the use of RNA-seq and RT-qPCR for transcriptomics studies has been widely reported and well-established for over a decade. There is no need to substantiate its benefits over older technologies and the inclusion of such paragraph adds little, if nothing, to the overall content of the Introduction. Please just refer the techniques used for assessing the metabolic differences between the CP and BP breeds.

Line 98-99: Were the RNA-seq muscle samples introduced in any sort of DNA/RNA conservation buffer such as RNAlater?

Line 109: Please define the meaning of L, a* and b* color phenotypes.

Line 123: Please add additional information of steps followed before Trizol homogenization. What was the method used to sample muscle tissue? Did you employ liquid nitrogen-based grinding in order to avoid any thawing of the sample during processing? Did you employ mechanic homogenization of the tissue samples? As a fibrous tissue, skeletal muscle needs to be properly grinded and homogenized by chemical or mechanical techniques in order to allow proper cell disruption and RNA extraction.

Line 125: NanoDrop does not provide any estimation of RNA integrity, but rather, an assessment of RNA concentration and possible presence or additional contaminants in the solution. RNA integrity should be assessed by calculating RIN values with a Bioanalyzer or equivalent equipment. Did you perform such analyses?

Lines 131-138: There are several relevant points missing here. Did you perform technical replicates for the RT-qPCR analyses? What was the scientific reasoning of using GAPDH as housekeeping gene? Which software did you use to process Ct values and normalize expression profiles from RT-qPCR reaction? Why didn’t you check additional housekeeping genes to assess their performance compared to GAPDH? What was the normalization procedure used? There are important pieces of information missing here. As it is, this section is not acceptable.

Lines 142-145: I assume that v91 refers to Ensembl version release. In that case, the current Ensembl release is v102, not v91, which was released back in December 2017.

Which library preparation kit was employed? Which sequencing method was used? Did you perform single-end or paired-end sequencing? Which software was used for trimming and quality-check of the reads? Which were the specifications for STAR alignment? Also, please properly cite both EdgeR and SeqMonk softwares and provide more details on which EdgeR specifications were used for DE analyses.

Lines 145-146: This last sentence of the paragraph adds no useful information. Please remove.

Lines 155-158: Please use correct terminology when defining the estimated statistics using a t-test approach. T-test analysis is used to assess differences between the mean of two groups, and normally Welch’s modification is used to control for differences in the variance between the groups. As it is, it seems that differences in the variance were assessed using a t-test, which is not, by any means, the case here.

Table 2: The use of star (*) notation for reflecting P-value significance should not be used as a substitute for reporting the actual P-values obtained in the analyses, but rather as a complement or graphical aid for better understanding the magnitude of the significance. Please show the P-values that were obtained in the analyses. As it is, Table 2 is not acceptable.

Table 3: Same comment as reported for Table 2 applies here.

Line 197: This claim is not supported by any means with your data. Please remove or clarify that you only provide a measurement of OCR phenotype and that mitochondrial concentration was not measured.

Figure 1: Same comment as reported for Table 1 and 2. Please report the actual P-values obtained.

Figure 2: Again, please show the P-values, not just star notation.

Table 4: Same comment as reported for Table 1 and 2 and Figures 1 and 2 regarding P-values. Also, given the small number of animals analyzed (8 BP x 7 CP), the estimates for phenotype distribution should not be considered as representative but rather as a small non-representative sample of the cohort of animals in each breed scheme. This reduced number of analyzed animals surely have a relevant impact in the accuracy of the analyzed fatty acids, and more prominently on those showing residual concentration. The observed significance for some of those fatty acids with very reduced concentration between the two groups are more probably due to type I (alpha) error induced by the small N number. Authors should be especially careful when interpreting residual differences in low concentrated fatty acids, given the very limited sample size employed in the study.

Lines 247-263: The authors report to have found 384 DE genes in their analyses, but no Table with FC or log2FC, Pvalue, FDR and expression values is reported. This should be included, at least, as a Supplementary Table. Please provide a Table reporting each of the 384 mentioned DE genes, with their corresponding FC, Pvalue, FDR and mean expression values in both groups.

Figure 3: The volcano plot should represent all genes included in the DE analyses. How many of them were used? Did you perform any sort of filtering to remove non-expressed or residually expressed genes? As it is shown, only DE genes

Figures 4 and 5: These Figures are incorrectly presented, analyzed and reported. The number of genes involved in a given GO term is irrelevant if no significance in the form of P-value or corrected P-value is reported. Also, the magnitude in X axis should report the significance of each term, rather than the number of genes involved from the list of DE genes. It is absolutely not clear which statistic was used to assess the significance of the reported GO terms according to DAVID database. Please report these analyses as Supplementary Table. Also, please modify Figures 4 and 5 adding a correct interpretation of the results. As they are in the manuscript, they are not acceptable.

Line 286: There is no such Table 7, but rather a Figure 7.

Figure 7: Why SLCA67 and FADS1 do not show significant differences represented by star notation? Also, there is no such “related gene expression” in your data, but rather a relative gene expression taking the CP group as reference (given the fact that all their expression values are normalized to 1, and BP group is reported as relative to the CP group expression profile for each gene). The fact that authors report 1 as baseline reference for CP group, reflects they did not perform log2 transformation of relative Ct normalized data, which might provide a better estimation of the relative differences between groups.

Author Response

Thanks for the comments.

  1. Lines 14-26: This section contains several flawed grammatical constructions and repetitions which should be fixed and reoriented. I suggest to rewrite and carefully consider the general claims that are put throughout the Manuscript.

Response: It has been rewritten.

  1. Lines 27-44: While I strongly suggest the authors not to overcrowd the abstract with acronyms, the use of these and their previous definition must be generalized for all of them used. For instance, the meaning of a*, h, SFA, MUFA and PUFA are not detailed. Please either do not use acronyms in the abstract, or apply a constant stable rule to define their corresponding meaning prior to their inclusion in the text.

Response: The SFA, MUFA and PUFA are detailed. The a* and h (hue) are commonsensible indexes in meat color characteristics. Could I keep it as a* and h?

  1. Lines 45-46: “Breeds” and “Gene Expression” are probably useless and not meaningful as keywords given the content of the Manuscript.

Response: They have been deleted.

  1. Line 54: Please introduce a grammatically correct form to introduce Dransfield et al. reference. As put in the text, it is grammatically linked to WHC, which is obviously not the case.

Response: It has been changed.

  1. Lines 56-57:  Please remove “eating” for defining meat quality. Also, the grammatically correct form of this sentence would be: “However, the effects of genotype on meat quality are generally higher than external feed conditions”.

Response: It has been changed, and “eating” has been removed.

  1. Line 57-58: Please rephrase this sentence.

Response: The sentence was rewritten.        

  1. Line 71-80: At present days, the use of RNA-seq and RT-qPCR for transcriptomics studies has been widely reported and well-established for over a decade. There is no need to substantiate its benefits over older technologies and the inclusion of such paragraph adds little, if nothing, to the overall content of the Introduction. Please just refer the techniques used for assessing the metabolic differences between the CP and BP breeds.

Response: This part has been revised, and the unimportant information has been removed.

  1. Line 98-99: Were the RNA-seq muscle samples introduced in any sort of DNA/RNA conservation buffer such as RNAlater?

Response: We used TRIzol Reagent and the method was described in the marital and method part. This information has been added to the RNA sequencing section of Material and Methods

  1. Line 109: Please define the meaning of L, a* and b* color phenotypes.

Response: The meaning of these indexes was given.

  1. Line 123: Please add additional information of steps followed before Trizol homogenization. What was the method used to sample muscle tissue? Did you employ liquid nitrogen-based grinding in order to avoid any thawing of the sample during processing? Did you employ mechanic homogenization of the tissue samples? As a fibrous tissue, skeletal muscle needs to be properly grinded and homogenized by chemical or mechanical techniques in order to allow proper cell disruption and RNA extraction.

Response: The muscle sampling method was mentioned in the 2.2 Carcass characteristics and sampling part. The samples were put into liquid nitrogen directly in order to avoid any thawing of the sample during processing. We employed mechanic homogenization of the tissue samples when treated them with Trizol reagent, and we added this step in the 2.5 RNA extraction and cDNA synthesis part.

  1. Line 125: NanoDrop does not provide any estimation of RNA integrity, but rather, an assessment of RNA concentration and possible presence or additional contaminants in the solution. RNA integrity should be assessed by calculating RIN values with a Bioanalyzer or equivalent equipment. Did you perform such analyses?

Response: Sorry for making the expression mistake. We only use the NanoDrop to measure the concentration and contaminants of the RNA. The integrity has been changed to concentration.

  1. Lines 131-138: There are several relevant points missing here. Did you perform technical replicates for the RT-qPCR analyses? What was the scientific reasoning of using GAPDH as housekeeping gene? Which software did you use to process Ct values and normalize expression profiles from RT-qPCR reaction? Why didn’t you check additional housekeeping genes to assess their performance compared to GAPDH? What was the normalization procedure used? There are important pieces of information missing here. As it is, this section is not acceptable.

Response: GAPDH is a traditional reference gene and widely used. Li, et al. 2011 indicated GAPDH is applicable as a reference gene in all pig tissues, and this citation has been inserted to the manuscript. A recent publication also used GAPDH as the reference gene in pig muscle stem cell study (Choi, et al. 2020. https://www.ncbi.nlm.nih.gov/pmc/articles/PMC7372987/#__ffn_sectitle). Other modifications have been conducted in this section

13. Lines 142-145: I assume that v91 refers to Ensembl version release. In that case, the current Ensembl release is v102, not v91, which was released back in December 2017.

Which library preparation kit was employed? Which sequencing method was used? Did you perform single-end or paired-end sequencing? Which software was used for trimming and quality-check of the reads? Which were the specifications for STAR alignment? Also, please properly cite both EdgeR and SeqMonk softwares and provide more details on which EdgeR specifications were used for DE analyses.

Response: The detail has been added to the article with supplementary Table and Figure. This study and the RNAseq analysis were performed in 2018 when v91 was the updated version. This was run as a BaseSpace app and the only option is the species you want to align the data to since filtering and adapter trimming was performed using the FastQC app in BaseSpace. EdgeR was run as a package in SeqMonk (SeqMonk has several methods for performing differential expression analysis and runs R in the background for many of these), so we don’t have any other citation to provide for this.

  1. Lines 145-146: This last sentence of the paragraph adds no useful information. Please remove.

Response: It has been removed.

  1. Lines 155-158: Please use correct terminology when defining the estimated statistics using a t-test approach. T-test analysis is used to assess differences between the mean of two groups, and normally Welch’s modification is used to control for differences in the variance between the groups. As it is, it seems that differences in the variance were assessed using a t-test, which is not, by any means, the case here.

Response: Sorry for the wrong expression. We used T-test to assess differences between the means of two groups. This part has been revised.

  1. Table 2: The use of star (*) notation for reflecting P-value significance should not be used as a substitute for reporting the actual P-values obtained in the analyses, but rather as a complement or graphical aid for better understanding the magnitude of the significance. Please show the P-values that were obtained in the analyses. As it is, Table 2 is not acceptable.

Response: The p-value has been added.

  1. Table 3: Same comment as reported for Table 2 applies here.

Response: The p-value has been added.

  1. Line 197: This claim is not supported by any means with your data. Please remove or clarify that you only provide a measurement of OCR phenotype and that mitochondrial concentration was not measured.

Response: I am sorry that we do not have the data, but the previous study could support the theory. The reference citied could support that.

  1. Figure 1: Same comment as reported for Table 1 and 2. Please report the actual P-values obtained.

Response: The p-value has been added.

  1. Figure 2: Again, please show the P-values, not just star notation.

Response: The p-value has been added.

  1. Table 4: Same comment as reported for Table 1 and 2 and Figures 1 and 2 regarding P-values. Also, given the small number of animals analyzed (8 BP x 7 CP), the estimates for phenotype distribution should not be considered as representative but rather as a small non-representative sample of the cohort of animals in each breed scheme. This reduced number of analyzed animals surely have a relevant impact in the accuracy of the analyzed fatty acids, and more prominently on those showing residual concentration. The observed significance for some of those fatty acids with very reduced concentration between the two groups are more probably due to type I (alpha) error induced by the small N number. Authors should be especially careful when interpreting residual differences in low concentrated fatty acids, given the very limited sample size employed in the study.

Response: The p-value has been added.

  1. Lines 247-263: The authors report to have found 384 DE genes in their analyses, but no Table with FC or log2FC, P value, FDR and expression values is reported. This should be included, at least, as a Supplementary Table. Please provide a Table reporting each of the 384 mentioned DE genes, with their corresponding FC, P value, FDR and mean expression values in both groups.

Response: The Supplementary Table has been provided.

  1. Figure 3: The volcano plot should represent all genes included in the DE analyses. How many of them were used? Did you perform any sort of filtering to remove non-expressed or residually expressed genes? As it is shown, only DE genes

Response: The figure 3 showed that there was a total of 384 differentially expressed genes found between these two breeds, in which 201 are highly differentially expressed (log2 Fold change ≥1 or ≤ -1; P-value <0.05). Compared with CP, BP had 75 up-regulated and 126 down-regulated genes.

  1. Figures 4 and 5: These Figures are incorrectly presented, analyzed and reported. The number of genes involved in a given GO term is irrelevant if no significance in the form of P-value or corrected P-value is reported. Also, the magnitude in X axis should report the significance of each term, rather than the number of genes involved from the list of DE genes. It is absolutely not clear which statistic was used to assess the significance of the reported GO terms according to DAVID database. Please report these analyses as Supplementary Table. Also, please modify Figures 4 and 5 adding a correct interpretation of the results. As they are in the manuscript, they are not acceptable.

Response: Figure 4 and 5 are just a classification for the genes that are highly expressed in BP or CP, respectively. There is no p-value.

  1. Figure 7: Why SLCA67 and FADS1 do not show significant differences represented by star notation? Also, there is no such “related gene expression” in your data, but rather a relative gene expression taking the CP group as reference (given the fact that all their expression values are normalized to 1, and BP group is reported as relative to the CP group expression profile for each gene). The fact that authors report 1 as baseline reference for CP group, reflects they did not perform log2 transformation of relative Ct normalized data, which might provide a better estimation of the relative differences between groups.

Response: There is no significant differences between two groups for SLCA67 and FADS1. Thank you for correcting my expression mistake. The “related” has been changed to “relative”. The p-value of all gene figure has been added.

Round 2

Reviewer 3 Report

  Dear Authors, I have gone through your revised Manuscript and, although partially improved, there are still several issues that must be addressed before any further consideration of the the current Manuscript for publication.   In the first round of revisions I pointed that:   "Second, phenotype-based comparison with such limited sample of animals is far from accurate and representative of the actual phenotype distribution across the population, and inferring significant differences with such limited N is quite risky, at best."   Authors seemed to ignore such relevant comment regarding the power of their phenotypic comparisons in their response letter. Yet, this is a very relevant problem of the Manuscript and should be stated in the body of the discussion, at least. Using 7 x 8 animals to infer phenotype differences applicable to both breed populations is very limited in terms of the reproducibility of the results Furthermore, we do not know what was the criteria for selecting such specific 7 and 8 animals for the experiment. Maybe they were specifically selected based on phenotype differences, thus rendering significant results afterwards. Such limited number of replication is a red alarm when attempting to extrapolate any of the results reported in this Manuscript to a generalized recommendation of using BP over CP pigs for animal production. Please add, at least, some reference to this issue in the text, so as readers are aware of the power limitations of the study.   More importantly, analyses of RNA-seq and Gene Enrichment data still seems flawed and wrong assumptions and presentation of the results are provided. I strongly suggest the Authors to carefully revise the methodology that was implemented for generating the RNA-seq data, as it contains, in my opinion, design errors that are spread downstream the analyses, making any discussion based upon such results unreliable or doubtful. More appropriately, I suggest to remove this part of the analyses if no good experimental design can be provided or justified. qPCR data seems, however, reliable and correctly executed, despite several problems with data presentation that are outlined in the following lines.   In summary, I strongly suggest Authors to take their time to carefully revise their Manuscript, carefully revise the methods and experimental design implemented and, if appropriate, remove and reform any of the sections that have been marked as flawed or inconsistent. Given the data presented, I would suggest to reduce the content of the Manuscript and to resubmit as a Short Communication focused on qPCR results and phenotype data (despite the limited power) in this or any other journal related to the Animal production field. I therefore suggest major revision of the Manuscript, again. Please take your time and do not resubmit similar results without carefully considering my comments. It is not my intention to undermine the Author's work but to highlight the problems of the Manuscript and to ensure a proper publication with reliable data. I will not consider this Manuscript further if another round of revisions require still Major modifications. If that would be the case, I will definitely Reject the Manuscript. My apologies for any possible inconvenience.     Major general comments:     Abstract   lines Please add the P-values obtained in scientific notation and use P-value, not just P. Please do not refer to P < 0.05   line 38: Please, as already requested, add the meaning of a* (redness or red to blue as put in the Manuscript) and h (hue angle) to the acronyms. These are common meat quality parameters but the general reader will benefit from these clarifications.   line 42: Please specify that the 201 DE genes were identified using the FC and FDR thresholds as reported in line 280-281.     Introduction   line 84: The RT-qPCR is not a verification of the RNA-seq results, but rather an additional method to support results obtained by RNA-seq. RT-qPCR is subjected to its own flaws and inconsistencies and should not be taken as a gold-standard for verification or confirmation of NGS results, although sadly highly referred as such.   line 85: Which reasons? I do not think that the term "reasons" is the adequate word to use according to the phrase.     Methods   line 140: It is not clear why you selected these specific 8 genes for RT-qPCR. According to RNA-seq results these mRNAs showed varied significance levels and do not seem to conform to any specific filtering. Were they selected based on their metabolic relevance for meat quality? You add some info regarding this in lines 314-332, but some clarification should also be added here.   line 142: Although this might be true, the proper method to implement when selecting housekeeping genes for RT-qPCR studies is to select some of the most commonly used housekeeping genes (3-4) in your tissue and species and test their comparative performance to assess whether they conform or not to be used in your experimental conditions. Sticking to one single widely used example is limiting and almost certainly a source of bias in your experiments. Please use proper RT-qPCR methodology in future studies.   line 147-148: Reactive conditions and primers used for RT-qPCR are definitely not presented in Table 3. Neither in Table 1, where only primers are depicted (see following suggestion).   line 151: I suggest to put Table 1 to Supplementary material. Also, binding site within the mRNA RefSeq transcript and amplicon size should be reported, as conforming with MIQE guidelines (https://academic.oup.com/clinchem/article/55/4/611/5631762).   line 160: When you say that "three biological pools were made for each breed", do you mean that RNA extracts for the 8 and 7 BP and CP pigs were pooled in triplicate BEFORE library prep and sequencing? If this was the case, you lost any putative individual gene expression variation, and your results from RNA-seq experiments are based on comparing two replicates composed by merged RNA pools from 7-8 different animals. This deeply weakens your results regarding DE genes in the muscle and make them unreliable, as the observed variation is derived from technical bias when pooling and sequencing, rather than from biological variance derived from the 7-8 analyzed animals, which was lost in the RNA pooling before indexing and sequencing. If that was the case, you are only comparing one pool against the other, despite the triple replication, which does not account for biological variance, but rather for technical variance during sequencing.   line 165: Why did you use single-end sequencing for mRNA quantification? Single-end sequencing is best suited for small RNAs, but not advisable for mRNAs with the current technologies available.   line 166-167: Please add references and/or web links to the tools used.   line 170-172: The correct link to cite Seqmonk v 1.37 is: http://www.bioinformatics.babraham.ac.uk/projects/seqmonk/. Also, please update your software tools for future research projects.   line 172: Please cite edgeR tool properly: https://academic.oup.com/bioinformatics/article/26/1/139/182458. Also, always refer to q-value or FDR when describing significance values after multiple testing correction. Using FD (I assume you mean False Discovery) is not correct). Also, you seem to have used some sort of FC filtering criteria, given your DE results and Figure 3. Please clarify which was the FC or log2FC filter to consider a gene as DE.   line 176: Method Paragraphs 2.8 and 2.9 would be better placed after Method 2.4 and before DE analyses. Please reorder.     Results   line 197: Table 1. Please add the star (*) symbol meaning to the Table caption or remove.   line 213-218: This section is poorly described and substantiated based on references added. Please carefully explain your differences and use proper references including further explanation of why those previously reported results are in accordance or add any sort of valuable information to your results.   line 227: As previously said in first round of revisions, this you simply do not know, you do not have data supporting this claim. If you want to hypothesize that, please refer as this being a non-tested hypothesis that requires further research in your animal model comparison. Do not use the term "should", because you simply do not know, neither directly nor indirectly.   line 234-235: This is simply not supported by your data. You just described differences in OCR phenotype. Please stick to discussion based on data you have reported or remove. Moreover, you linked OCR with mitochondrial concentration, not with mitochondrial higher metabolism. This entire paragraph should be carefully reconsidered.   line 272: Table 2. Please add the star (*) symbol meaning to the Table caption or remove.   line 281: Regarding Table S3, please carefully reformat. For reporting DE results, we certainly do not need to see columns B to E. Also, we do need to see both P-values (not reported) and q-values (column F). Column G I simply do not know what these values are. Columns H and I should be placed as columns A and B, followed by FC or log2FC column (choose one between columns V or W), P-values and FDR. Moreover, it is not clear which group is CP and which is BP, as you used a B and W terminology for identifying both groups. These are the minimum set of data that must appear in any table listing DE genes. Also, expression values reported in columns N to U are quite confusing. Which units are they expressed in? Only columns T and U are needed in any case. Furthermore, as previously mentioned, the fact that you seem to have used a pooled triplicate for each group is not a proper design for DE analyses. One should expect to see individual non-pooled measurements for each analyzed animal. Pooling RNA extracts will remove any kind of transcriptional variance in your data, leading to a focus on technical variance from sequencing runs, which is not reflecting real transcriptomic differences among animals and among groups. If my assumptions are correct, I strongly suggest to remove RNA-seq analyses from the paper, as they are not reliable as a measure of differences between CP and BP pigs, given that, in reality, you are only contrasting 1 BP vs 1 CP replicate capturing real transcriptomic differences. You simply lost your replication power pooling your RNA extracts, and your triplicates are a measure of sequencing and alignment bias.   Figures 4, 5 and 6: Again, if you used DAVID to perform Pathway Enrichment and GO Enrichment analyses, you surely provided your list of putative DE genes to this software, which returned a list of enriched Pathways and GO terms, their significance, P-values and q-values and the number of genes and genes involved in each of the highlighted terms. If you want to report these enriched pathways and GO terms, please add their significance. The number of genes in each term is irrelevant if you do not provide the significance of the enrichment. These Figures are meaningless. Also, you made no attempt to discuss the results you report in them. I suggest to remove them, perform proper Pathway and GO enrichment analyses and discuss the results you obtained, if any.   line 314-332: I strongly suggest to focus on discussing the results obtained for qPCR analyses, as RNA-seq methodology seems flawed, although some of the results might be in accordance with those obtained with qPCR. In line 318, you refer that FOS shows higher expression in CP compared with BP, but in RNA-seq FOS has a positive FC, similar to SLPI gene, which has higher expression in BP (see Suppl Table 3). So your qPCR results do not always comply with RNA-seq as you previously mentioned in line 316.   line 383: The sequenced reads should be uploaded to publicly available repositories such as NCBI SRA.          

Minor grammatical corrections:

Summary   line 20: "compare to..." >> compared with   line 22: "the carcass, meat quality traits..." >> the carcass and meat quality traits.     Abstract   line 35: "was used..." >> were used   line 42???     Introduction   line 62: "phenotype [4]." >> phenotypes [4].   line 69-70: Please rephrase the "long and deep-bodied" part.   line 71: "the carcass, meat quality traits..." >> the carcass and meat quality traits.   line 75-81: There is a repetition error here. Also, please remove "and it has been widely used as it can provide a complete comprehension of genotype items [7,8]". This adds nothing to the Manuscript.     Methods   line 104: "muscles of longissimus dorsi" >> muscle samples of longissimus dorsi.   line 104: "One piece of samples was stored" >> LD muscle samples were snap frozen at -80ºC in liquid nitrogen   line 112: "a polypropylene bag then stored" >> a polypropylene bag and then stored   line 116: "was calculated by" >> were calculated using   line 119: "Ten grams minced" >> Ten grams of minced   line 125: "composition was analysis" >> composition was analysed   line 135: "The RNA were used" >> The same RNA extracts were used   line 142: "GAPDH has been widely used as the housekeeping gene in most of the swine tissues [14]." >> GAPDH gene has been widely used as housekeeping normalizer [14], and was thus selected as reference in the present study.      Results   line 209: "that BP is higher than CP" >> meaning that IMF of BP is higher than that of CP pigs.   line 227: "It was reported than muscle" >> It was reported that.

Author Response

Thanks for the valuable comments.

Cover letter

  1. Abstract   lines Please add the P-values obtained in scientific notation and use P-value, not just P. Please do not refer to P < 0.05

Response: It shows the significant differences already, and it is concise. Besides, most of the article in this journal are presented like this way in abstract.

  1. line 38: Please, as already requested, add the meaning of a* (redness or red to blue as put in the Manuscript) and h (hue angle) to the acronyms. These are common meat quality parameters but the general reader will benefit from these clarifications. 

Response: It has been added.

  1. line 42: Please specify that the 201 DE genes were identified using the FC and FDR thresholds as reported in line 280-281. 

Response: It has been added.

  1. Introduction   line 84: The RT-qPCR is not a verification of the RNA-seq results, but rather an additional method to support results obtained by RNA-seq. RT-qPCR is subjected to its own flaws and inconsistencies and should not be taken as a gold-standard for verification or confirmation of NGS results, although sadly highly referred as such.

Response: It has been changed to support.

  1. line 85: Which reasons? I do not think that the term "reasons" is the adequate word to use according to the phrase.     

Response: This sentence has been revised.

  1. Methods   line 140: It is not clear why you selected these specific 8 genes for RT-qPCR. According to RNA-seq results these mRNAs showed varied significance levels and do not seem to conform to any specific filtering. Were they selected based on their metabolic relevance for meat quality? You add some info regarding this in lines 314-332, but some clarification should also be added here. 

Response: The reason has been written in lines 319-320. These genes are related to lipid deposition or metabolism.

  1. line 142: Although this might be true, the proper method to implement when selecting housekeeping genes for RT-qPCR studies is to select some of the most commonly used housekeeping genes (3-4) in your tissue and species and test their comparative performance to assess whether they conform or not to be used in your experimental conditions. Sticking to one single widely used example is limiting and almost certainly a source of bias in your experiments. Please use proper RT-qPCR methodology in future studies.

Response: Thank you for your suggestion.

  1. line 147-148: Reactive conditions and primers used for RT-qPCR are definitely not presented in Table 3. Neither in Table 1, where only primers are depicted (see following suggestion).   line 151: I suggest to put Table 1 to Supplementary material. Also, binding site within the mRNA RefSeq transcript and amplicon size should be reported, as conforming with MIQE guidelines (https://academic.oup.com/clinchem/article/55/4/611/5631762).

Response: These have been added.

  1. line 160: When you say that "three biological pools were made for each breed", do you mean that RNA extracts for the 8 and 7 BP and CP pigs were pooled in triplicate BEFORE library prep and sequencing? If this was the case, you lost any putative individual gene expression variation, and your results from RNA-seq experiments are based on comparing two replicates composed by merged RNA pools from 7-8 different animals. This deeply weakens your results regarding DE genes in the muscle and make them unreliable, as the observed variation is derived from technical bias when pooling and sequencing, rather than from biological variance derived from the 7-8 analyzed animals, which was lost in the RNA pooling before indexing and sequencing. If that was the case, you are only comparing one pool against the other, despite the triple replication, which does not account for biological variance, but rather for technical variance during sequencing.

Response: The biological pool is made by mixing 2~3 samples together so the RNA sequencing procedure can be performed as 3v3 in a run. It is a limitation of the method and the equipment that was available. It is commonly used in animal research compared to medical science, because the genetic background in livestock animals is much more unified compared to the human genome. But we appreciate your suggestion, in the future, we will consider a better way to analyze the data.

  1. line 165: Why did you use single-end sequencing for mRNA quantification? Single-end sequencing is best suited for small RNAs, but not advisable for mRNAs with the current technologies available. 

Response: The reason was the limitation of the availability of the method and equipment.

  1. line 166-167: Please add references and/or web links to the tools used. 

Response: It has been added.

  1. line 170-172: The correct link to cite Seqmonk v 1.37 is: http://www.bioinformatics.babraham.ac.uk/projects/seqmonk/. Also, please update your software tools for future research projects.

Response: It has been added.

  1. line 172: Please cite edgeR tool properly: https://academic.oup.com/bioinformatics/article/26/1/139/182458. Also, always refer to q-value or FDR when describing significance values after multiple testing correction. Using FD (I assume you mean False Discovery) is not correct). Also, you seem to have used some sort of FC filtering criteria, given your DE results and Figure 3. Please clarify which was the FC or log2FC filter to consider a gene as DE. 

Response: The citation has been added. The filter to consider a gene as DE has already been described in the results and discussion part. log2 Fold change ≥1 or ≤ -1; P-value <0.05

  1. line 176: Method Paragraphs 2.8 and 2.9 would be better placed after Method 2.4 and before DE analyses. Please reorder.

Response: It has been reordered.

  1. Results   line 197: Table 1. Please add the star (*) symbol meaning to the Table caption or remove.

Response: It had been already removed in the first revision.

  1.  line 213-218: This section is poorly described and substantiated based on references added. Please carefully explain your differences and use proper references including further explanation of why those previously reported results are in accordance or add any sort of valuable information to your results. 

Response: The references just support our data.

  1. line 227: As previously said in first round of revisions, this you simply do not know, you do not have data supporting this claim. If you want to hypothesize that, please refer as this being a non-tested hypothesis that requires further research in your animal model comparison. Do not use the term "should", because you simply do not know, neither directly nor indirectly. 

Response: This sentence has been removed.

  1. line 234-235: This is simply not supported by your data. You just described differences in OCR phenotype. Please stick to discussion based on data you have reported or remove. Moreover, you linked OCR with mitochondrial concentration, not with mitochondrial higher metabolism. This entire paragraph should be carefully reconsidered.

Response: This sentence has been removed.

  1. line 272: Table 2. Please add the star (*) symbol meaning to the Table caption or remove.

Response: It is just the correction label.

  1. line 281: Regarding Table S3, please carefully reformat. For reporting DE results, we certainly do not need to see columns B to E. Also, we do need to see both P-values (not reported) and q-values (column F). Column G I simply do not know what these values are. Columns H and I should be placed as columns A and B, followed by FC or log2FC column (choose one between columns V or W), P-values and FDR. Moreover, it is not clear which group is CP and which is BP, as you used a B and W terminology for identifying both groups. These are the minimum set of data that must appear in any table listing DE genes. Also, expression values reported in columns N to U are quite confusing. Which units are they expressed in? Only columns T and U are needed in any case. Furthermore, as previously mentioned, the fact that you seem to have used a pooled triplicate for each group is not a proper design for DE analyses. One should expect to see individual non-pooled measurements for each analyzed animal. Pooling RNA extracts will remove any kind of transcriptional variance in your data, leading to a focus on technical variance from sequencing runs, which is not reflecting real transcriptomic differences among animals and among groups. If my assumptions are correct, I strongly suggest to remove RNA-seq analyses from the paper, as they are not reliable as a measure of differences between CP and BP pigs, given that, in reality, you are only contrasting 1 BP vs 1 CP replicate capturing real transcriptomic differences. You simply lost your replication power pooling your RNA extracts, and your triplicates are a measure of sequencing and alignment bias.   Figures 4, 5 and 6: Again, if you used DAVID to perform Pathway Enrichment and GO Enrichment analyses, you surely provided your list of putative DE genes to this software, which returned a list of enriched Pathways and GO terms, their significance, P-values and q-values and the number of genes and genes involved in each of the highlighted terms. If you want to report these enriched pathways and GO terms, please add their significance. The number of genes in each term is irrelevant if you do not provide the significance of the enrichment. These Figures are meaningless. Also, you made no attempt to discuss the results you report in them. I suggest to remove them, perform proper Pathway and GO enrichment analyses and discuss the results you obtained, if any.

Response: Legend has been added to explain the B and W issue. As I mentioned before, the pooling method is common in animal studies with very similar genetic backgrounds in livestock animals. The object of this study is to compare the genetic differences between Large Black pigs and commercial pigs. So it is hard to delete the RNA sequencing and only discuss the growth performance and qPCR data.

  1. line 314-332: I strongly suggest to focus on discussing the results obtained for qPCR analyses, as RNA-seq methodology seems flawed, although some of the results might be in accordance with those obtained with qPCR. In line 318, you refer that FOS shows higher expression in CP compared with BP, but in RNA-seq FOS has a positive FC, similar to SLPI gene, which has higher expression in BP (see Suppl Table 3). So your qPCR results do not always comply with RNA-seq as you previously mentioned in line 316.

Response: A word ‘partly’ has been added.

  1.  line 383: The sequenced reads should be uploaded to publicly available repositories such as NCBI SRA. 

Response: We can do it but will require extra time to submit and publish. We only have five days for this round of review. If it is mandatory and the editor agrees, we can do it with extra time.

  1. Minor grammatical corrections:

Summary   line 20: "compare to..." >> compared with   line 22: "the carcass, meat quality traits..." >> the carcass and meat quality traits.     Abstract   line 35: "was used..." >> were used   line 42???     Introduction   line 62: "phenotype [4]." >> phenotypes [4].   line 69-70: Please rephrase the "long and deep-bodied" part.   line 71: "the carcass, meat quality traits..." >> the carcass and meat quality traits.   line 75-81: There is a repetition error here. Also, please remove "and it has been widely used as it can provide a complete comprehension of genotype items [7,8]". This adds nothing to the Manuscript.     Methods   line 104: "muscles of longissimus dorsi" >> muscle samples of longissimus dorsi.   line 104: "One piece of samples was stored" >> LD muscle samples were snap frozen at -80ºC in liquid nitrogen   line 112: "a polypropylene bag then stored" >> a polypropylene bag and then stored   line 116: "was calculated by" >> were calculated using   line 119: "Ten grams minced" >> Ten grams of minced   line 125: "composition was analysis" >> composition was analysed   line 135: "The RNA were used" >> The same RNA extracts were used   line 142: "GAPDH has been widely used as the housekeeping gene in most of the swine tissues [14]." >> GAPDH gene has been widely used as housekeeping normalizer [14], and was thus selected as reference in the present study.      Results   line 209: "that BP is higher than CP" >> meaning that IMF of BP is higher than that of CP pigs.   line 227: "It was reported than muscle" >> It was reported that.

Response: All of them have been revised.

Round 3

Reviewer 3 Report

My comments have been uploaded in the corresponding document called "Review3".

Author Response

Major comments
Line 127: Authors should provide information about the crossbred commercial line used in the
study. Which breeds are involved in this commercial porcine line?

Re: The line is PIC 29 dam × PIC 380 boar

Line 235-236: Table 3 does not report reactive conditions for PCR or primers. This information
is provided in Table 1, but please do not use the term “reactive conditions” for referring to Table
1 contents, as these are just primers data. Also, in previous revisions I requested to provide the
binding position of used primers, which were not detailed in Table 1. Please provide the binding
position of primers and specify if these were exon-exon spanning or not. Moreover, I suggest to
put the Table 1 as Supplementary, which was a previous ignored request.

Re: This has been added to the 2.8 section. Polymerase activation and DNA denaturation at 95℃ for 3 mins, followed by 40 cycles of denaturation at 95℃ for 15s, annealing/extension at 55℃ for 30s. The binding position info has been added to Table 1. The statement of the primers spanning introns has been added, too.

Line 245-247: There is no information regarding which pigs are number 1, 2, 3, 4, 5, 6, 7 or 8 for
each group. I suggest to detail phenotypic data for each animal used in a Supplementary Table
and to clearly identify them with these numbers. Otherwise, please rephrase.

Re: It has been rephrased. No specific pig number shows.

Line 269: Please add reference to the Star alignment software:
https://academic.oup.com/bioinformatics/article/29/1/15/272537?login=true. And please, do not
put it as a link but rather as an additional reference.

RE: This reference has been added.  Dobin, A., Davis, C. A., Schlesinger, F., Drenkow, J., Zaleski, C., Jha, S., ... & Gingeras, T. R. (2013). STAR: ultrafast universal RNA-seq aligner. Bioinformatics29(1), 15-21.

Line 273: This error reporting “p<0.05 after FD correction” was already spotted in previous
revisions. Please, do not put this same error a third time. When referring to multiple testing
correction significance, always use q or q-value, not p or p-value, as this refers to nominal P-value
before multiple correction. I do not know what FD means, if you mean false discovery rate for
multiple testing correction, the correct term to use is FDR. Please use the proper terminology. I
will not accept the Manuscript in any circumstance when these unacceptable errors are still in the
text.

Re: EdgeR applies the Benjamini-Hochberg method on the p-values, to control the false discovery rate (FDR) and calls the output a “FDR adjusted p-value”. The different methods for calculating “q values” use a Bayesian approach to estimate the positive false discovery rate. here is a link to an article that describes different FDR methods: https://bmcbioinformatics.biomedcentral.com/articles/10.1186/s12859-018-2081-x

Line 274: Please put the edgeR paper as an additional reference, not as a link.

RE: The reference has been added : Robinson, M. D., McCarthy, D. J., & Smyth, G. K. (2010). edgeR: a Bioconductor package for differential expression analysis of digital gene expression data. Bioinformatics26(1), 139-140.

Line 277: Please move this section before the 2.7 section.

Re: Usually the statistical method is put at the last in the materials and methods section in animal science papers. But since you insisted, it has been moved.

Regarding results reported in Tables 2, 3 and 4, as already requested, please add a phrase
regarding the limitations of the study for determining fatty acids composition and other
phenotypes given the limited number of animals analyzed for each group. As authors only used 7
and 8 pigs from both analyzed groups, all reported significant phenotypic differences might be
subjected to strong bias. Authors and readers should not generalize the obtained results to a breedwide interpretation unless further analyses using a higher number of animals are performed. Also,
differences reported in fatty acids with very low concentration should be taken as highly
unreliable and subjected to further detailed analyses. Again, please refer to this limitation and its
implications in the Manuscript. I will not accept this Manuscript until such issue has been clearly
stated.

Re: the phrase of a statement has been added to the results and discussion section after 3.5 section.

Line 405: Again, you are not using P-value (P) < 0.05 for reporting DE results, but rather q-value
(FDR) < 0.05 (Column F in Table S3). Please, use the proper terminology, this is totally
unacceptable, misleading and confusing.

Re: Similarly, to the question above.

As previously highlighted in other revisions, results regarding Figures 4, 5 and 6 are wrong and
misleading. If you want to report Pathway Enrichment and GO terms for your DE genes, you
must report enriched terms with their corresponding significance at P-value and q-value after
multiple testing correction (FDR or Bonferroni). These results are unacceptable, please change or
remove. I will not accept this Manuscript until such errors have been corrected.

Re: We have changed the title of Figures 4, 5, and 6. We actually report the numbers of the pathways enriched in each group based on the GO analysis p-value <0.05.

Line 549: A Supplementary Table with results for DE analyses on all considered genes must be
provided. Sequencing files must be uploaded to NCBI SRA database. Raw Ct data for qPCR
analyses must also be provided in the form of Supplementary Table. Also, a Supplementary Table
with phenotype data on a per-animal basis is highly advisable. Making this data available is easy
and required in order to ensure the feasibility of the analyses and reported results.
Supplementary Table S3: Please, as already requested, report DE results in the following form >>
Gene ID, Gene name, FC, P-value, FDR. Columns T and U can also be included. Please remove
the rest of columns, they are useless. I will not accept the Manuscript until the Authors make the
effort to properly report their RNA-seq results.

Re: The database uploading to the NCBI SRA database has been done. And it is awaiting approval. It will be released once approved. Here is the link: http://www.ncbi.nlm.nih.gov/bioproject/689570 

Minor comments
Line 108: Remove “the world”

Line 110: This sentence (“In contrast, the Large Black pigs (BP) are British native pigs with long
and deep-bodied”), is incorrect. What is long? the body, the carcass?

Line 116: Please change “The RNA-seq which is widely used to provide a complete
comprehension of genotype items have been applied” >> The RNA-seq methodology, which is
widely used to provide a comprehensive transcriptomic profile of analyzed tissues, has been
applied

Line 117: Please change “On the base of RNA-seq, functional analysis of Gene Ontology (GO)
biological process (BP) can be used to highlight the genes association and the networks of relevant
biological pathways” >> On the basis of RNA-seq results, functional analyses of Gene Ontology
(GO) and biological processes were used to highlight enriched relevant metabolic pathways.
Please remove the biological process (BP) acronym as it interferes with the already used BP for
Large Black pigs.

Line 119: Please remove refs 9 and 10 and reorder appropriately.

Line 120-122: Please remove this last sentence.

Line 125: Please remove “Before the initiation of research”

Line 135: Please change “off feed” >> fasted

Line 180: Please change “sample were bloomed for 20 minutes before experiment” >> samples

were bloomed for 20 minutes before further analyses

Line 188: Please change “After filtered” >> After filtering

Line 494: Duplicated “the”. Please correct.

Re: All minor comments have been corrected in the manuscript.